# The Marine Alga *Sargassum horneri* Is a Functional Food with High Bioactivity

Masayoshi Yamaguchi

Cancer Biology Program, University of Hawaii Cancer Center, University of Hawaii at Manoa, 701 Ilalo Street, Hawaii, HI 96813, USA; yamamasa11555@yahoo.co.jp

**Abstract:** Functional food factors can play a preventive and therapeutic role in several human diseases. The marine alga *Sargassum horneri* (*S. horneri*) has restorative effects in several types of metabolic disorders, including osteoporosis, diabetes, inflammatory conditions, and cancer cell growth. Osteoporosis is widely recognized as a major public health problem. Bone loss associated with ageing and diabetic states was prevented through the intake of bioactive compounds from *S. horneri* water extract in vivo by stimulating osteoblastic bone formation and inhibiting osteoclastic bone resorption in vitro. The intake of *S. horneri* water extract was found to have preventive effects on diabetic findings with an increase in serum glucose and lipid components. Furthermore, the *S. horneri* component has been shown to suppress adipogenesis from rat bone marrow cells and inflammatory conditions in vitro. Notably, the growth of bone metastatic human breast cancer MDA-MB-231 cells, which induce bone loss with osteolytic effects, was suppressed through culturing with the *S. horneri* water extract component in vitro. The *S. horneri* component, which has a molecular weight of less than 1000, was found to suppress the activation of NF-κB signaling by tumor necrosis factor-α, a cytokine associated with inflammation, in osteoblastic cells and macrophage RAW264.7 cells in vitro, suggesting a molecular mechanism. The bioactive component of *S. horneri* may play a multifunctional role in the prevention and treatment of metabolic disorders. This review outlines the advanced knowledge of the biological activity of the aqueous extract components of *S. horneri* and discusses the development of health supplements using this material.

**Keywords:** *Sargassum horneri*; osteoporosis; diabetes; obesity; cancer; NF-κB signaling; dietary supplement





## 1. Introduction

Functional food factors can play a preventive and therapeutic role in several human diseases. The knowledge that dietary factors regulate biological functions provides a wealth of information about food and health. In recent years, the marine alga *Sargassum horneri* (*S. horneri*) has been shown to have restorative effects in several types of metabolic disorders, including osteoporosis, diabetes, inflammatory conditions, and cancer cell growth. Osteoporosis, the bone loss associated with ageing, is widely recognized as a major public health problem. Therefore, it may be of great interest to elucidate the role of food-derived factors in the regulation of bone metabolism and to investigate the prevention and repair of osteoporosis [1–3].

Bone tissue is formed by the destruction of old bone by osteoclasts, which break down bone minerals, and the formation of new bone tissue by osteoblasts, which make new bone. The bone then continues to become flexible and elastic [4–6], which is called bone remodeling [4]. It has a mechanism to hold it in place. When this balance is disturbed by ageing or many pathological conditions, bone mass decreases, leading to osteoporosis. This bone disease makes people susceptible to fractures, causes them to be bedridden with fractures, disrupts their daily lives, and accelerates their death. It can play an important role in maintaining health and helping to prevent and repair disease. The incidence of bone

disease, including osteoporosis, has been increasing annually with the recent increase in the ageing population, and there has been much interest in its prevention.

The author has a strong interest in elucidating the role of food-derived factors in the regulation of bone metabolism and research related to the prevention and repair of osteoporosis [1–3]. Alongside fisheries researchers with morphological characteristics, we collected seaweeds from the sea in Shizuoka Prefecture (Shimoda City, Japan), such as Undaria pinnatifida, *Sargassum horneri*, Eisenia bicyclis, and Cryptonemia schmitziana. Gelidium amansii and Ulva ulvaceae from Lake Hamana (Shizuoka, Japan) were used to study the effects of their extracts on bone calcium content [4]. As a result, we initially discovered that, among edible algae, the water-extracted components of *Sargassum horneri* (Japanese name, Akamoku) have a strong effect on increasing bone calcium levels in rat femoral tissue in vitro [7]. After that, the components of several seaweeds have been reported to express osteogenic effects [8–14]. *S. horneri* is found along the coasts of Japan and China, growing from winter to spring on rocky areas of the ocean floor where the waves are relatively calm. This seaweed grows rapidly and becomes entangled in fishing nets and ship propellers, so it is rarely used. Although it has been used as food in some areas, much of it has been discarded into the environment. This seaweed is a brown alga belonging to the kelp, mozuku (Japanese), and wakame (Japanese; *Undaria pinnatifida*) families. It has just the right amount of sliminess and a unique chewy texture, and has been used as a delicious and healthy seaweed since ancient times in areas such as Akita, Iwate, Toyama, and the Noto Peninsula in Japan. It was eaten raw. In addition, *S. horneri* (Japanese name, Akamoku) is known to contain significantly higher amounts of nutritional components than other seaweeds, such as carotene, vitamin C, vitamin B2, dietary fiber, minerals, and trace elements.

Subsequently, we found that in addition to promoting bone mass, the water extract components of *S. horneri* have the effects of ameliorating hyperglycemia and hyperlipidemia in diabetes in vivo, and suppressing the formation of fat cells associated with obesity both ex vivo and in vivo. [15]. It has been shown that these effects are based on a mechanism by which the *S. horneri* components suppress the expression and activity of nuclear factor kappa B (NF-κB), an intracellular information transduction molecule that is important for the expression of the above pathological conditions [16]. Thus, the water extract component of *S. horneri* was found to be effective in preventing and ameliorating a variety of pathological conditions, and it is expected to be effective as a traditional Chinese herbal medicine with a variety of physiological activities that help improve health.

This mini review outlines the advanced knowledge of the biological activity of the aqueous extract components of *S. horneri* and discusses the development of health supplements using this material.

## 2. *S. horneri* Component May Prevent Osteoporosis

### 2.1. Unique Anabolic Effect of S. horneri Component in Bone Tissue

Seaweed extract components were prepared to clarify the biological effects of various seaweeds on bone tissue in vitro. *S. horneri* extract was proven to increase the amount of calcium in the femoral tissue in a dose-dependent manner [7].

The experiment used a lyophilized sample of the extract dissolved in the purified distilled water of various seaweeds. Rats were given a 5% aqueous solution of aqueous extracts of wakame, red moss, arame, occidental ibis, maculata, and sea lettuce orally for 7 consecutive days. Among the different seaweeds, the intake of *S. horneri* aqueous extract uniquely increased the amount of calcium in femoral metaphyseal tissue (spongy bone tissue) in a dose-dependent manner [7]. In addition, the activity of alkaline phosphatase, an enzyme that promotes bone mineralization, and the amount of DNA (evaluated as an index of the number of cells in bone tissue) were shown to be significantly increased [17]. Such an effect of *S. horneri* extract was also induced in the diaphyseal (cortical) tissues of rat femurs. The anabolic effects of the *S. horneri* extract component were based on the enhancement of bone protein synthesis in bone cells [7,17]. The aqueous extract

component of *S. horneri* may contain a bioactive factor that enhances osteoblast function, which promotes bone formation [7,17]. Brown algae contain several active compounds such as fucoidan, fucosterol and fucoxanthin. We have confirmed that these substances do not have an anabolic effect on bone formation.

It is noteworthy that the aqueous extract component of *S. horneri* has also been shown to suppress bone resorption through the action of osteoclasts causing osteolysis [18]. When the diaphyseal and metaphyseal tissues of rat femurs were cultured in the presence of bone resorption-promoting factors such as parathyroid hormone and prostaglandin E2, bone calcium content decreased [18]. These effects were almost completely suppressed in the presence of the aqueous extract component of *S. horneri*. Thus, the aqueous extract of *S. horneri* was found to have an inhibitory effect on bone resorption in vitro.

## 2.2. Characterisation of the Active Ingredient in S. horneri Aqueous Extract

An attempt was made to identify the active compounds in the aqueous extract of *S. horneri*. Firstly, the molecular weight of the active compounds in *S. horneri* water extract was determined using membrane fractionation [19]. The compounds that suppress bone resorption have been shown to have a molecular weight of 50,000 or more. To further separate the active components, the active fraction of the Sephadex G-25 (Sigma-Aldrich, St. Louis, MO, USA) column elution of the red mosquito extract was separated by high performance liquid chromatography (Asahipack GF-310HQ (Resonac, New York, NY, USA)) and GS-220HQ (Resonac, New York, NY, USA). The final fraction was found to contain several new and known substances with molecular weights (MW) of approximately 350–400 [19]. Finally, we found the presence of 4 chemicals in the *S. horneri* components (less than MW 3000) by analysis using a liquid chromatography–mass spectrophotometry system (LCMS-IT-TOF; Shimadzu, Kyoto, Japan). These chemicals were identified as 1,3,5-tris(oxolan-2-ylmethyl)-1,3,5-triazinane (MW 339), 5-phenyl-2-[2-(5-phenyltetrazol-2-ethyl)] tetrazole (MW 318), 3-(hexadecylamine)propane-1,2-diol (MW 316), and 2-(2-hydroxyethyl-tridecylamino)ethanol (MW 288). These chemicals may affect osteoblastogenesis and/or osteoclastogenesis. The combination of these compounds has a more potent anabolic effect on bone than either component alone. The manifestation of this potent combined effect may be important in the expression of multi-functionality.

## 2.3. S. horneri Component Has an Inhibitory Effect on the Activation of the NF-κB Signalling Pathway

Endogenous tumor necrosis factor-alpha (TNF-α) decreases peak bone mass and inhibits osteoblastic Smad activation through NF-κB [20–22]. The *S. horneri* component has been shown to have anabolic effects on bone tissue [7,19]. To elucidate the mechanistic property, we used the *S. horneri* components with molecular weights below 3000. The inflammatory cytokine TNF-α plays a central role in the molecular and cellular mechanisms of skeletal pathology [20,21]. We focused on the involvement of TNF-α in the expression of the anabolic effects of *S. horneri* on bone tissue. Inhibition of NF-κB, which is associated with TNF-α signaling, promotes osteoblastic bone formation and suppresses osteoclastic bone resorption in vitro. Notably, the aqueous extract component of *S. horneri* was found to block the impairment of osteoblast function by TNF-α and the control of osteoclast activation by the receptor activator of NF-κB (RANK) ligand (RANKL) [16]. This result suggests that the suppression of TNF-α signaling plays a critical role in exerting the anabolic effect of the *S. horneri* extract component on bone. We found that there are several new and known chemical substances with molecular weights between 200 and 400 in the aqueous extract components of *S. horneri* (MW3000 or less) [19]. These molecules may contribute to the inhibition of NF-κB signaling activation. Mechanistically, this inhibition may be important in exerting the bone anabolic effects of the *S. horneri* aqueous extract component.

The cellular mechanism by which the *S. horneri* component exerts an anabolic effect on bone metabolism, leading to increased bone mass, is described in Figure 1.

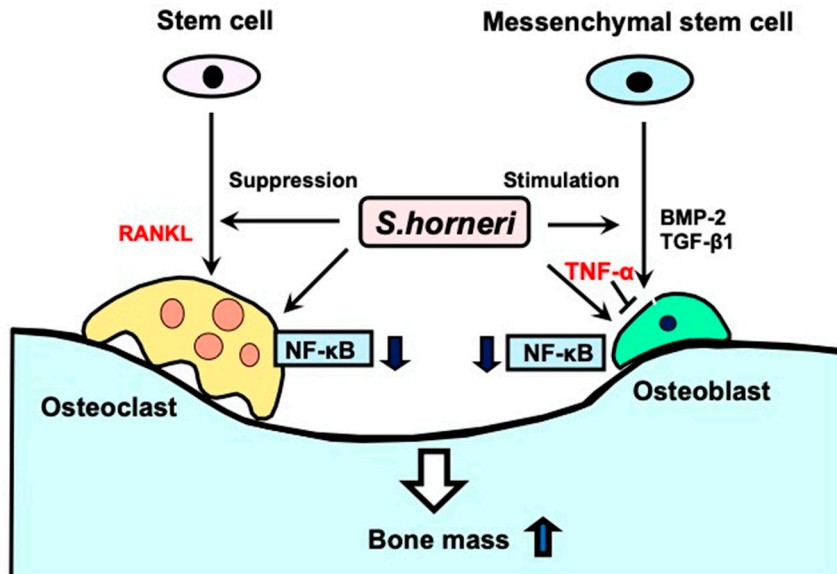

**Figure 1.** The cellular mechanism by which the *S. horneri* component exerts its osteogenic effects. Osteoclasts are differentiated from stem cells. RANKL stimulates osteoclastogenesis. Osteoblastic cells are differentiated from bone marrow mesenchymal stem cells and are stimulated by bone growth factors (TGF-β1 and BMP-2). The *S. horneri* component stimulates osteoblastic bone formation and suppresses osteoclastic bone resorption, thereby increasing bone mass. The *S. horneri* component suppresses TNF-α- and RANKL-enhanced NF-κB activation in osteoblasts and osteoclasts, suggesting a possible molecular mechanism for the osteogenic effects of the *S. horneri* component. In addition, bone marrow mesenchymal stem cells are differentiated into adipocytes. The *S. horneri* component suppresses adipogenesis from bone marrow mesenchymal stem cells. This may lead to the prevention of obesity and associated bone loss.

## 2.4. Involvement of S. horneri Water Extract Component in Osteoporosis Prevention

As mentioned above, *S. horneri* aqueous extract has been shown to promote bone formation and inhibit bone resorption [7,15–19]. In addition, we investigated its effect on promoting bone constituents and preventing osteoporosis in growing and ageing rats [23]. First, growing (4-week-old male) and ageing (50-week-old female) rats were given 2.5, 5 and 10 mg of *S. horneri* water extract (lyophilized product) per 100 g of rat body weight once a day [16]. After continuous oral administration for 7 and 14 days, changes in bone constituents (calcium content, alkaline phosphatase activity and DNA content) in the diaphyseal (bone) and metaphyseal (cancellous bone) tissues of the femur were examined [23]. Significant increases in all bone components, including calcium content, alkaline phosphatase activity, and DNA content, were found in the diaphyseal and metaphyseal tissues of growing and ageing rats [23]. Thus, it has been demonstrated that the intake of *S. horneri* components has the effect of promoting bone mass during both growth and ageing.

Osteoporosis with bone loss as a complication of diabetes has attracted clinical attention as an intractable disease [24,25]. Streptozotocin (STZ) is used as a model animal for diabetes because it destroys the pancreas and induces type 1 diabetes due to impaired insulin secretion [26]. In STZ-treated rats, bone constituents (calcium, alkaline phosphatase activity and DNA content) in the diaphysis and metaphysis of the femur were significantly reduced [15]. Notably, STZ-induced bone loss was significantly suppressed by continuous oral administration of *S. horneri* extract for 14 and 21 days in vivo [15]. Thus, the water extract component of *S. horneri* may have an improved effect on diabetic osteoporosis and may be useful in the supplemental prevention of osteoporosis.

*2.5. Complementary Effects of S. horneri Water Extract Component in Humans*

Based on the above knowledge, a new functional food material was developed using *S. horneri* water extract as the active ingredient to prevent osteoporosis. The material of the final product was named Hormax OT, after the name *S. horneri*. Note that OT is an abbreviation for osteon (meaning bone). Generally, healthy volunteers (24 men and women, 11 to 13 in each group) took 1 tablet (containing 300 mg of Formax OT) or 3 tablets (containing 900 mg of Formax OT) a day for 4 or 8 weeks [27].

This study was analyzed by measuring the serum concentration of bone turnover markers developed for the clinical evaluation of osteoporosis. Note that a starch tablet was used as a placebo. Bone turnover markers included bone-type alkaline phosphatase [28] and osteocalcin, markers of bone formation (proteins specifically produced by osteoblasts) [29], tartrate-resistant acid phosphatase, a marker of bone resorption (a protein produced by osteoclasts) [30], and type 1 collagen degradation product N-telopeptide (a specific degradation product of type 1 collagen protein present in bone tissue during bone resorption) [31].

Specifically, Formax OT was found to significantly reduce tartrate-resistant acid phosphatase, a serum marker of bone resorption, and suppress bone resorption (bone mineral dissolution) [24]. It was concluded that *S. horneri* extract constituents exert their bone resorption inhibiting effect early in the intake and then promote bone formation, thus helping to maintain bone mass.

In addition, general blood biochemistry tests and blood cell counts did not change significantly after taking 1 or 3 tablets of Formax daily for 8 weeks, and no toxicity was observed [27]. The functional material "Formax OT", which was first developed at Maruhachi Muramatsu Co., Ltd. (Shizuoka, Japan), is a powdered version of the water extract component of *S. horneri*, which has been shown to be effective in humans and may be effective in preventing osteoporosis.

## 3. *S. horneri* Water Extract Component Has a Preventive Effect on Diabetes

Whether the *S. horneri* component ameliorates diabetic states has been investigated using model rats treated with STZ, which destroys the pancreas and induces type 1 diabetes due to impaired insulin secretion [8]. In this rat model, suppressed body weight gain, elevated serum glucose and triglyceride concentrations, and elevated serum calcium and inorganic phosphorus concentrations were observed [15]. These diabetes-related fluctuations were significantly improved through continuous oral administration of the *S. horneri* water extract component for 14 and 21 days [15]. This result suggests that there are factors in the *S. horneri* water extract component that exert anti-diabetic effects and that its intake is effective in improving diabetic conditions. In addition, it is believed that there are two factors in the *S. horneri* water extract component, one that improves diabetic conditions and one that repairs diabetic osteoporosis, as described previously [15]. However, the same factor may be effective in improving both conditions, suggesting that they may be related.

## 4. *S. horneri* Extract Component Suppresses Adipogenesis in Bone Marrow Cells

Bone marrow stem cells are pluripotent and differentiate into osteoblasts, chondrocytes, cardiomyocytes, and adipocytes [32–34]. We have also found that factors in the aqueous extract component of *S. horneri* suppress the formation of adipocytes from bone marrow cells [35]. The question of whether bone marrow stem cells differentiate into adipocytes or osteoblasts has attracted much interest in the progression of osteoporosis. Therefore, we investigated the formation of adipocytes by culturing mouse bone marrow cells in a culture medium containing a component of *S. horneri* water extract (molecular weight less than 3000) in vitro [35]. We found that the *S. horneri* water extract component enhanced the insulin-stimulated differentiation of bone marrow stem cells into osteoblasts and suppressed the formation of adipocytes [35]. *S. horneri* extract also has inhibitory effects on lipid metabolism in 3T3-L1 adipocytes in vitro [36]. Thus, the *S. horneri* component may have the effect of suppressing the differentiation of bone marrow cells into adipocytes.

TNF-$\alpha$ secreted by mature adipocytes may be involved in the pathogenesis of obesity-induced osteoporosis [37,38]. In addition, the factors in the aqueous extract component of *S. horneri* were found to block the inhibition of osteoblast function and osteoclast hyperplasia caused by the activation of intracellular NF-$\kappa$B signaling stimulated by TNF-$\alpha$ [16]. Note that TNF-$\alpha$ is upregulated in bone marrow adipocytes and mature adipocytes and is an important factor in causing inflammation and insulin resistance [35]. The *S. horneri* component has been shown to improve the status of type 1 diabetes [15]. The *S. horneri* component may also be useful in preventing type 2 diabetes, which is associated with obesity and insulin resistance.

## 5. Suppressive Effects of the *S. horneri* Component on the Inflammatory State Associated with TNF-$\alpha$ in Various Cell Types

The *S. horneri* component has been shown to exert suppressive effects on the inflammatory state of various cell types, including macrophages, retinal cells, and dermal fibroblasts [39–51]. In this mini review, the author has focused on inflammatory macrophages. The *S. horneri* component with 70% EtOH extracts, such as phenolics and flavonoids, has been shown to have excellent anti-inflammatory and antioxidant activities [39]. Levels of several cytokines, including prostaglandin E2 (PGE2), TNF-$\alpha$, and interleukin (IL)-6, which mediate pro-inflammatory effects, were found to be reduced by treatment with *S. horneri* extract [39]. *S. horneri* extract has been shown to downregulate pro-inflammatory cytokines, PGE2 and nitric oxide secretion by blocking the downstream activation of Toll-like receptor (TLR)-mediated NF-$\kappa$B and mitogen-activated kinase (MAPK) phosphorylation [40,50]. Furthermore, *S. horneri* extract inhibited chronic lung inflammation induced by particulate matter by blocking TLR/NF-$\kappa$B/MAPK signaling in lung macrophages [42]. TNF-$\alpha$, which stimulates TLR signaling, is important in the development of inflammatory diseases and cancer, and is a contributing factor in a variety of diseases. It may be critical as a common mechanism by which the *S. horneri* component has suppressive effects on the activation of NF-$\kappa$B signaling.

## 6. Suppressive Effects of the *S. horneri* Component on Cancer Cells

Among various types of cancer, bone metastasis is a very serious and deadly disease. Breast cancer cells metastasize to bone, and approximately 70% of patients with advanced breast cancer will develop bone metastases [52–57]. Breast cancer promotes the formation of osteoclasts by secreting osteoporotic cytokines, including TNF-$\alpha$ [58,59]. Treatment includes drugs and antibodies that inhibit the action of RANKL, a cytokine that activates osteoclast function leading to bone resorption [20,21]. We found that the aqueous extract component of *S. horneri* stimulates osteoblastic bone formation and osteoclastic bone resorption, leading to bone loss [7,16,17], and investigated whether the *S. horneri* component affects breast cancer cells.

MDA-MB-231 human breast cancer cells have bone metastatic potential and lack the expression of estrogen receptor alpha, progesterone, and epidermal growth factor receptor 2. MDA-MB-231 triple negative breast cancer cells are well established as a model of human breast cancer that is difficult to treat with drugs [52,53,57,59]. We investigated whether the *S. horneri* component would be useful in controlling breast cancer. In the culture of human breast cancer MDA-MB-231 cells, the *S. horneri* component with a molecular weight of less than 3000 was found to suppress proliferation with cell cycle arrest and stimulate apoptotic cell death of MDA-MB-231 cells, resulting in a reduced numbers of cancer cells [60]. This study may suggest that the *S. horneri* compound is a useful tool for cancer prevention and therapy without side effects.

In addition, a polysaccharide fraction obtained from *S. horneri* by hot water extraction was shown to inhibit the growth of human colon cancer DLD cells in a dose-dependent manner by inducing apoptosis of DLD cells [61]. The polysaccharide component obtained from *S. horneri* caused the accumulation of cells in G0/G1 and S phase and affected the expression of apoptosis-associated genes such as Bcl-2 and Bax [62]. This may explain the

growth inhibition of DLD cells. These studies suggest that sulphate content and molecular weight may influence antioxidant and antitumor activities. Thus, the *S. horneri* component has been shown to have anticancer activity. Further studies are anticipated.

### 7. Conclusions

Research into the biological activities of *S. horneri* has been fruitful, with the discovery in 2001 that it has osteogenic effects [7]. This is popular because it helps prevent osteoporosis, which leads to fractures and is common in an ageing population. This disease is common in bed-ridden people and is considered a socially important disease wherein prevention is important. The aqueous extract component of *S. horneri* has been shown to have an osteogenic effect by stimulating osteoblastic bone formation and suppressing osteoclastic bone resorption, leading to the prevention of age-related bone loss. Among the many edible seaweeds, *S. horneri* was found to have a unique osteogenic effect. Subsequently, the *S. horneri* component was found to have suppressive effects against diabetes, adipogenesis, obesity, inflammation, and cancer cell growth. Mechanistically, the *S. horneri* component was shown to suppress the TNF-$\alpha$-activated transcription factor NF-$\kappa$B signaling, which is associated with several diseases. *S. horneri* may be a functional food with multifunctional bioactivities. In addition, these bioactive components may be needed to identify the detailed chemical structure and associated biological activity. This study may lead to the development of new pharmaceutical drugs. Thus, the *S. horneri* marine algae component has a variety of physiological activities and is expected to be used as a material for supplements with Chinese herbal medicine effects.

**Funding:** This research received no external funding.

**Institutional Review Board Statement:** This article does not include any human or animal studies conducted by any of the authors.

**Informed Consent Statement:** Not applicable.

**Data Availability Statement:** The datasets used in this study are available from the respective authors upon reasonable request.

**Conflicts of Interest:** The author declares no conflicts of interest.

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
