# Peer review of "The Marine Alga Sargassum horneri Is a Functional Food with High Bioactivity"

_nutraceuticals, doi:10.3390/nutraceuticals4020012_

Round 1

Reviewer 1 Report

Comments and Suggestions for Authors

Overall, this review is too focused on potential bioactivity of s. horneri in osteoporosis. even though this manuscript also describe about other diseases, it is not covered as good as osteoporosis section. Figure is good and easy to understand. The knowledge gap is also already mentioned. Unfortunately, for a review manuscript, this has quite lot of self citation (in reference list there are some Yamaguchi, M. name, and also the author often mentioned about his/her own research)

Comments on the Quality of English Language

The English used in this manuscript is understandable, but extensive editing of English language is required for a better manuscript.

Author Response

Comments of Reviewer 1

Overall, this review is too focused on potential bioactivity of s. horneri in osteoporosis. even though this manuscript also describe about other diseases, it is not covered as good as osteoporosis section. Figure is good and easy to understand. The knowledge gap is also already mentioned. Unfortunately, for a review manuscript, this has quite lot of self citation (in reference list there are some Yamaguchi, M. name, and also the author often mentioned about his/her own research)

Reply of the author:

Reply of the author:

 Thank you very much for your kind comments.

We searched through PubMed using the term of S.horneri and its biological effects. We could not find out the findings of S. horneri’s biological effects besides its bone effects. Therefore, this author introduced our findings, especially bone effects.

However, we found the findings of inflammation and cancer reported by other authors. We introduced these findings in this article.

In the end, the elucidation of horneri’s biological effects as functional food needs further studies. This author introduced the current findings of horneri in this mini-review. The reader may have an interest in the horneri’s biological effects.

Research on this aspect will be developed in feature studies.

In addition. this manuscript was edited by native English researchers and by using English software for grammatical and editing.

Reviewer 2 Report

Comments and Suggestions for Authors

Yamaguchi suggested that the bioactive component of S. horneri may play a multi-functional role in health maintenace. 

Despite the current manuscript beding well summarized, there are some concers as below;

1. In the introduction, the discrimination techniques that can distinguish between S. horneri and other brown alge is should be described with the correct reference.

2. Please refrain from repeated citations more than three times, such as  reference 7, 9, 10, 11, 14 etc.

3. Basically, a large number of brown algae including S. horneri contain various active substances such as fucoidan, fucosterol, and fucoxanthin etc. So, please serve more information in detail in the each subresult section.

4. If author serve picture for explaining the structure of an active ingredient or chemical, it will help to undrstand for readers in the paragraph 2.2.

5. It is necessary to correct the color or size of the font, i.e. Figure 1.

6. If author serve summerized table in human study of S. horneri effects, it will help to undrstand for readers (the paragraph 2.5).

7. In recent studies, S. horneri has anti-inflammatory effects in the various type of cells such as macrophages, retinal cells, and dermal fibroblasts. So, please serve more information with the correct references in subresult section 5 "Suppressive effects of the S. horneri component on the inflammatory state".

8. It must moify typo errors, and repetitive words.

Comments on the Quality of English Language

Dear Editor

Please see the comments and suggestions.

I hope author solves all of these concerns one by one. 

Author Response

Comments and Suggestions for Authors

Yamaguchi suggested that the bioactive component of S. horneri may play a multi-functional role in health maintenace. Despite the current manuscript beding well summarized, there are some concers as below;

  1. In the introduction, the discrimination techniques that can distinguish between S. horneri and other brown alge is should be described with the correct reference.

Reply of author:  We could not find out reference to distinguish between S.horneri and other brown algae. In general, researchers compare their difference with morphologically. Fisheries researchers can easily distinguish.  This described in the section of Introduction (third paragraph, line 3).

  1. Please refrain from repeated citations more than three times, such as  reference 7, 9, 10, 11, 14 etc.

Reply of author: This was corrected in manuscript.

  1. Basically, a large number of brown algae including S. horneri contain various active substances such as fucoidan, fucosterol, and fucoxanthin etc. So, please serve more information in detail in the each subresult section.

Reply of author: Thank you very much for your kind suggestion. Brown algae  contain various active substances such as fucoidan, fucosterol, and fucoxanthin. We confirmed that these substances did not have an anabolic effect on bone formation. This was described in the section of 2.1 (secondparagraph, line 11-14).

  1. If author serve picture for explaining the structure of an active ingredient or chemical, it will help to undrstand for readers in the paragraph 2.2.

Reply of author: Thank you very much for your kind suggestion. We are under study and patentable. Therefore, we did not show an active ingredient or chemical. We think that the complexed ingredient may reveal a strong biological effect as described in this section.

  1. It is necessary to correct the color or size of the font, i.e. Figure 1.

Reply of author: Thank you very much for your kind suggestion. We revised in Figure 1.

  1. If author serve summerized table in human study of S. horneri effects, it will help to undrstand for readers (the paragraph 2.5).

Reply of author: Thank you very much for your kind suggestion. We put references. However, in the section of paragraph 2.5, the author revised manuscript with further modification to make it easy for the reader to understand.

  1. In recent studies, S. horneri has anti-inflammatory effects in the various type of cells such as macrophages, retinal cells, and dermal fibroblasts. So, please serve more information with the correct references in subresult section 5 "Suppressive effects of the S. horneri component on the inflammatory state".

Reply of author: Thank you very much for your kind suggestion. We selected the most suitable references in various studies. Especially, we focused macrophages. However, this author revised section 5.

  1. It must moify typo errors, and repetitive words.

Reply of author: Thank you very much for your kind suggestion. This was corrected in manuscript.

Round 2

Reviewer 1 Report

Comments and Suggestions for Authors

The author has confirmed the revision clearly and understandable. But It still has to be revised based on the previous comment. Please send the revised paper, not only the response about the overall comment.

Author Response

Comments of Reviewer 1

 Unfortunately, for a review manuscript, this has quite lot of self citation (in reference list there are some Yamaguchi, M. name, and also the author often mentioned about his/her own research)

  1. Reply of the author: The author added 17 papers of other authors in this manuscript. Self citations were 14 papers and total references are 64.

  1. The author checked your suggested manuscript pdf and revised:
  2. Title was changed: The marine algae Sargassum horneri is a functional food with potent bioactivity
  3. Is it important to write species name in italic in this entire manuscript:

Reply: S. horneri was changed italic.

  1. It is crucial to include the main gap in S. horneri bioactive material as supplementary food in abstract section. Also, in the abstract section it is important to elaborate the objectives or aims of this review manuscript; e.g what kind of bioactive would be exploring, disease, mechanism, etc.

Reply: The author revised for the comments of reviewer.

  1. It is also need to be written in abstract section

Reply: As indicated by the reviewer, the author included this in the abstract section.

  1. It is advisable to not too describe method that used in previous studies while reviewing the effect of S. horneri in bone tissue. It is enough to mention that s. horneri extract was proven to increase the amount of calcium in the femoral tissue in dose-dependent manner.

Reply: As indicated by the reviewer, the author revised.

  1. Is it too method-based review? and quite lot of self citation (also give attention in section 2.1). it is advisable to explore more about similar study from other previous studies.

Reply: Thank you very much for your kind suggestion. The author liked to introduce this method.

  1. The gap in knowledge has been identified

Reply: The author deleted this sentence.

  1. Reply: The describtion of repeated Reference 16 was deleted.

  1. What is the meaning of improving diabetes in this matter? do you mean s. horneri water extract could enhanced the diabetes possibility or what? please be clear about it

Reply: The author revised this sentence.

  1. This section is a good example to review. instead of focusing or mentioning on what method has been used, it should be elaborate more about the various results that has been gained by other studies.

Reply: The author added more references of other researchers as Refs.43-50.

Reviewer 2 Report

Comments and Suggestions for Authors

1. There are quite a few self-citations. It's better to explore more about similar studies from other previous studies.

2. The revised version still has typo errors and repetitive word.

Comments on the Quality of English Language

The revised version still has typo errors and repetitive word.

Author Response

Comment of Reviewer 2

  1. There are quite a few self-citations. It's better to explore more about similar studies from other previous studies.

Reply of the author: I added 17 papers of other authors in this manuscript. Self citations were 14 papers and total references are 64.

  1. The revised version still has typo errors and repetitive word.

Reply of the author: The author checked typo errors and corrected it.

Thank you very much.
